# Life Satisfaction in Employed Mothers of Children with Disabilities: The Importance of Personal, Family, Work, and Society Characteristics

**Matilda Nikolić Ivanišević** [1,*] [iD]**, Ana Slišković** [1] [iD]**, Jelena Ombla** [1] [iD]**, Andrea Tokić** [1] [iD] **and Theresa Brown** [2] [iD]

[1]   Department of Psychology, University of Zadar, 23000 Zadar, Croatia; aslavic@unizd.hr (A.S.); jlevac@unizd.hr (J.O.); apupic@unizd.hr (A.T.)
[2]   Department of Psychology and Counseling, Georgian Court University, Lakewood, NJ 08701, USA; tbrown@georgian.edu
*   Correspondence: mnikolic@unizd.hr

**Abstract:** The aim of this study was to determine the separate and joint contribution of individual, family, occupational, and social factors in explaining the life satisfaction of working mothers of children with developmental disabilities. Working mothers of children with disabilities participated in this study (N = 508). They completed an online questionnaire to measure factors from personal (optimism and personal strength), family (satisfaction with family finances, parental stress, number of children, and support from family members related to work), work (job demands, control, and support) and society domain (satisfaction with the healthcare, educational and welfare system). All of them were employed (at least part-time) but also, they all had at least one child with disabilities under 19 years of age whose degree of disability was officially determined. Regression analysis indicated that factors from personal, family (satisfaction with family finances, parental stress, and support from family members related to work), and societal domain (satisfaction with the healthcare system) predicted mothers' life satisfaction. Work-related variables did not. A comprehensive approach is very useful in studying the well-being of parents of children with disabilities. Future studies should also include fathers, as it is reasonable to assume that mothers and fathers differ in the influence of certain factors on their well-being. Considering the sample size and bias, these results have significant limitations in terms of generalizability.

**Keywords:** life satisfaction; employed mothers of children with disabilities; personal characteristics; family characteristics; work characteristics; society

## 1. Introduction

The birth of a child brings great changes for parents, and the birth of a child with developmental disabilities makes these changes and challenges even more pronounced [1]. Different child diagnoses bring some specific difficulties, both for the child and for the parents. However, there are many similarities between parents of children with developmental disabilities, regardless of the specific disability. The care of a child is usually long-term and accompanies the parents throughout their lives [2]. Parents feel a special responsibility for their child's development and want to understand and learn how best to support the child's development [3]. The issues parents face are largely determined by the child's age (e.g., expectations regarding the child's verbal development, and social, or motor skills), which requires constant parents' adjustment [4]. In addition, parents are forced to deal with increased financial costs [5]. Often, due to the increased care of the child, at least one parent withdraws from the labor market [6]. The list of similarities is far more extensive than the one above, but it is sufficient to conclude that parents of children with developmental disabilities face qualitatively and quantitatively greater demands compared to parents of children without disabilities, making them vulnerable to negative consequences, both for

themselves and ultimately for the family and the child itself. Reviewing the literature, there can be found a number of articles stating that parents' well-being is compromised, whether through life satisfaction, or mental and/or physical health [7–10]. It should be noted, however, that a review of the literature also shows wide variability on the issue of parents' adjustment. Some work focuses on the positive aspects of parenting children with disabilities, such as post-traumatic growth [11] or increased perceptions of parenting competence [12]. Obviously, in this "journey" from finding out the child's diagnosis to well-being, there are many factors that can have a protective or aggravating effect, and thus contribute to the variability of adjustment and well-being. These factors can come from different domains. Authors Ombla et al. [13] proposed a classification of factors into four domains: individual, family, work, and society.

### 1.1. Individual Domain

In a qualitative study [13] in which 25 working parents of children with developmental disabilities were interviewed, parents highlighted optimism, emotional stability, patience, and organization as factors that promote their well-being and help them balance between private and work roles. In this article, we tried to focus on some individual strengths/characteristics that help a person face increased demands and challenges. Optimism has been shown to predict higher overall well-being and lower parenting stress by other studies [14–16]. In the study [14], increased optimism was found to predict increased positive feelings and decreased negative feelings, even after controlling for child problem behavior and parenting stress. The authors concluded that optimism may impact how parents perceive their children with disabilities. In the area of studying the well-being of parents of children with disabilities, there are a relatively small number of studies comparing mothers and fathers. In one such longitudinal study [15], optimism was found to be a moderator of the relationship between children's behavior problems and parental well-being, primarily for mothers. When children's behavior problems were high, mothers who were less optimistic reported lower scores on measures of well-being than mothers who were more optimistic. Recently, researchers have also looked at some positive changes following parenting children with disabilities, and one of them is post-traumatic growth. It has been confirmed that a child's difficulties can bring positive changes in a parent's daily life, their well-being, and a change in their beliefs about themselves, the world, and the future [17,18].

### 1.2. Family Domain

Families with children with developmental disabilities make up an increasingly large proportion of households as a consequence of better care for newborns and the deinstitutionalization process. In the United States, for example, the number of noninstitutionalized children with developmental disabilities has doubled since the 1960s [19]. In Croatia, most children with developmental disabilities also live in the family: 97.5% of them [20].

Data on the prevalence of children with developmental disabilities vary depending on the method of data collection, the definition used, and cultural differences in the understanding of developmental disabilities [21]. According to the International Classification of Functioning, Disability and Health for Children and Youth [22], difficulties in child development include four main domains: body structures (e.g., the structure of the nervous, sensory, cardiovascular, and other body systems), body functions (e.g., sensory, mental, psychomotor, etc.), limitations in activities (e.g., walking, reading, learning) and participation (e.g., playing), as well as physical, social, and psychological environmental factors. The mentioned areas are also included in the Croatian national classification of children with developmental disabilities: "A child with developmental disabilities is a child who, due to physical, sensory, communication, language or intellectual disabilities, needs additional support for learning and development in order to achieve the best possible developmental outcome and social inclusion" [23] (p. 10). Estimates at the global level [24] suggest that there are between 2.5 and 10% of children with developmental disabilities in the population.

According to the data from the Croatian National Register of Persons with Disabilities [25], the number of children whose developmental difficulties are classified as disabilities (age 0–19) is 69,953, of which 42,910 are boys and 27,043 are girls, and their share in the general population of children is 9.4%.

Living with a child with developmental disabilities can have profound effects on all aspects of family life and bring many changes for family members and their relationships. One of the most important risk factors for parental well-being is the child's level of disability [26,27]. The number of children is also an important predictor of parental well-being. A greater number of children with difficulties in the family is associated with lower well-being [28], while the presence of a child/children without difficulties in the family is associated with higher well-being [26]. Caring for children with developmental disabilities means that parents must seek various forms of therapy and rehabilitation, which ultimately imposes additional financial costs on the family. As families are exposed to increased costs over a long period of time, this often causes them to slip into poverty over time [29,30]. The level of financial stress has been linked to health outcomes for parents [5]. Some studies suggest that parents of children with developmental disabilities are more likely to divorce [31], but other studies do not confirm these findings [32]. Research findings consistently show that this group of parents has higher levels of parenting stress compared to parents whose children have a normal developmental trajectory [33]. However, research also shows that resources such as social support, problem-focused coping, and positive affect help buffer the stress associated with parenting [34,35].

### 1.3. Work Domain

A particularly significant challenge for this group of parents is formal employment and balancing the demands of work and family roles. Parents find themselves in an unenviable situation where family expenses are increased, so the need for income and work is also increased. On the other hand, childcare is also increased, usually forcing one parent to withdraw from the labor market. Most often, this is the mother due to the gender-expected role as caregiver, which is reflected in higher unemployment rates or generally longer periods of unemployment compared to fathers [36,37]. The relationship between the working status of parents and the well-being of parents of children with disabilities is not unambiguously determined. For some parents of children with disabilities, employment may play a protective role in well-being because of the many functions that work has, but it may also be an additional source of stress that can lead to lower well-being [7]. Employed parents of children with disabilities face numerous challenges related to balancing the demands of work and family. The above challenges have been extensively researched in the general population, most notably through the concepts of work–family balance, work–family conflict, and work–family integration [38–44]. This topic is much less explored in the population of parents of children with disabilities. Authors Brown and Clark [45] reviewed 54 studies published on this topic and concluded that organizational factors that impact work–life balance are supervisor support, workplace policies, and organizational culture. In this research, we used Karasek's Job Demand–Control–Support model [41] to assess the relationship between workplace characteristics and employee well-being. According to this model, job demands are not the only thing that determines an employee's well-being, but it is important to consider these demands in the context of the control or autonomy that the employee has in his or her job, and in the context of the support provided by colleagues and/or supervisors. A higher level of control and a higher level of support have a positive effect on the well-being of the worker and can act as protective factors even in situations where work demands are extremely high [46,47]. Karasek's model is one of the most studied models of occupational stress since its appearance [47,48]. However, in reviewing the available literature, we did not encounter information that the model had been used in the context of parents of children with disabilities. We also considered this model appropriate because it includes the dimension of support, which has previously been shown to be an organizational factor important to worker well-being [45].

*1.4. Society*

Considering the increased demands that characterize the care of children with disabilities, their parents need the help (formal and informal) of the community, since these demands exceed the capacity of the family. The formal supports available to this vulnerable group and how they are implemented vary in each state, making cross-national comparisons difficult. In general, formal support is required through the health and education systems and in the exercise of various rights (e.g., extended maternity leave, various forms of financial support). In Croatia, these rights are prescribed by two laws [23,49]. The rights that parents and their children with developmental disabilities can exercise relate to three broad groups: the family's right to financial support (the right to personal disability allowance, the right to assistance and care allowance, the right to compensation for education-related transportation costs, the right to children's allowance), the parent's rights under the parental work system (the right to part-time work, the right to caregiver status), and the children's rights under the educational and rehabilitation system (the right to psychosocial support, early intervention services, the right to assistance with integration into educational programs) [50]. Exercising these rights is important for parents of children with developmental disabilities, as they primarily enable support relevant to the development of their children, but also shape the possibility of parents' employment. The shortcomings related to the social sphere in the Republic of Croatia can be divided into two broad categories. One is the inadequacy of the prescribed laws [50], and the other is the gap between the legal provisions and their application [51].

It is obvious that the well-being of parents of children with disabilities is influenced by numerous factors from different domains, and for working parents, the situation is even more complex. The aim of this study was to determine the separate and joint contribution of individual, family, occupational, and social factors in explaining the life satisfaction of working mothers of children with developmental disabilities. The individual characteristics studied were optimism and personal strength (one of the dimensions of posttraumatic growth). According to the literature [16,18], we expect these two variables to correlate positively with life satisfaction. Of the family factors, the number of children, parental stress, satisfaction with family finances, and social support from family members were tested. Parental stress is expected to be negatively correlated with life satisfaction, while all other correlations are expected to be positive. The assumption regarding the number of children is based on previous research [26] and the expected structure of the sample since more children usually mean having a child without difficulties. Of the estimated job characteristics, demands are expected to be negatively correlated with life satisfaction, and control and support are expected to be positively correlated with it. The social domain is measured as satisfaction with three systems: healthcare, education, and welfare. A positive correlation is expected between satisfaction with each of these systems and life satisfaction. Because this study is comprehensive, capturing predictors from four domains, the percentage of explained variance of life satisfaction is expected to be large. Each set of predictors from different domains is expected to contribute significantly to the percentage of explained variance in life satisfaction.

## 2. Materials and Methods

*2.1. Data and Sample*

A total of 508 working mothers who have children with developmental disabilities participated in this study. A convenient sample was used, and data were collected using an online questionnaire, after obtaining approval from the Ethics Committee of the Department of Psychology of the University of Zadar. All participants met three conditions: (1) they had a child with an objectively diagnosed developmental disability aged up to 19 years; (2) they were employed; and (3) had residence in the Republic of Croatia (the importance of factors from the social domain was also studied, the sample had to be homogeneous in terms of residence). The questionnaire was distributed through institutions that deal with children with disabilities and their parents such as kindergartens, schools, hospitals,

Croatian Welfare Institute. Some of the participants were reached via social networks and associations uniting parents of children with developmental disabilities. The structure, i.e., description of the sample is given in Table 1.

**Table 1.** Description of sample participants (N = 508).

| Characteristics | Descriptive Parameters |
|---|---|
| Age | *M* = 40.77 years, *SD* = 5.99, Min = 23 Max = 59 |
| Education | Elementary school (N = 4)<br>Secondary school (N = 213)<br>Bachelor's degree (N = 75)<br>Master's degree (N = 185)<br>Doctoral degree (N = 31) |
| Marital status | Extramarital relation (N = 54)<br>Married (N = 396)<br>Divorced (N = 42)<br>Single (N = 13)<br>Widow/er (N = 3) |
| Number of children | Mode = 2 (f = 243), Min = 1, Max = 6 |
| Number of children with disabilities | Mode = 1 (f = 462); Min = 1, Max = 4 |
| Age of children with disabilities | *M* = 9.19 years, *SD* = 4.86, Min = 0, Max = 19 |
| Degree of disability | 4th—most severe impairment (N = 185)<br>3rd—severe impairment (N = 139)<br>2nd—moderate impairment (N = 62)<br>1st—mild impairment (N = 31)<br>Do not know/not sure (N = 91) |
| Working hours | Full-time (N = 329)<br>Part-time (N = 172) |
| Employment sector | Public (N = 240)<br>Private (N = 268) |
| Partner's employment | Yes (N = 419)<br>No (N = 19)<br>Caregiver (N = 12) |

*2.2. Measures*

Life satisfaction, as the criterion variable, was assessed with the Life Satisfaction Scale [52], adapted version of the Satisfaction with Life Scale [53]. It consists of 5 items that measure global cognitive assessments of satisfaction with one's life. An example of particle is: "If I could live my life over again, I would change almost nothing". Participants are asked to indicate their level of agreement with each statement on a 7-point scale (1—strongly disagree to 7—strongly agree). The total score is the sum of the responses to all five statements, with a higher score representing greater satisfaction. Cronbach $\alpha$ is 0.89.

Optimism was assessed using the Optimism–Pessimism Scale [54], adapted version of the Optimism and Pessimism Scale [55]. This is a two-dimensional scale with a total of 14 items, 6 of which relate to optimism (the rest to pessimism), and they were used in this study. For each item, respondents rate the extent to which it applies to them on a scale from 1 to 5 (1—does not apply to me at all; 5—applies to me completely). Example of a particle: "I always look at things from the positive side". The total score is the sum of the scores of assessments on all particles, and a higher result indicates greater optimism. Cronbach $\alpha$ is 0.85.

*Personal strength* was assessed by the *Personal Strength subscale* from the Posttraumatic Growth Questionnaire [56], which is an adapted and validated version of the Posttraumatic Growth Inventory (PTGI) [57]. The instruction was adapted to the participants of this study, and it is emphasized that the changes they assess refer to the changes that occurred after

being confronted with the child's diagnosis. The Personal Strength subscale consists of 4 items (e.g., "I know I can cope better with difficulties"). Participants rate the extent to which they have experienced change on a 6-point scale (from 0 = I have experienced no change to 5 = I have experienced change to an extremely high degree). The total score is the sum of the ratings and ranges from 0 to 105 (for the entire questionnaire), but it is possible to calculate and use scores in individual growth areas. In general, a higher score indicates a higher level of posttraumatic growth overall or in a particular area of change. Cronbach $\alpha$ is 0.89.

*Parental stress* was assessed using the *Parental Stress Scale* [58], an adapted and validated version of Berry and Jones [59]. The scale captures satisfaction with fulfilling the parental role, positive and negative emotions associated with this role, and difficulties associated with parenting. The scale consists of 18 items, and respondents indicate the extent to which they agree or disagree with a particular statement on a Likert scale. Both positive (e.g., "I am happy in my role as a parent") and negative statements (e.g., "I am too burdened with the responsibilities of parenthood") are included. Responses range from 1 (strongly disagree) to 5 (strongly agree). Answers to some items are scored inversely. The final score is formed as a simple sum of the scale scores and can range from 18 to 90 so that a higher score on the scale indicates greater parental stress and vice versa. Cronbach $\alpha$ is 0.89.

*Social support from family members* was assessed in the context of work, and family members included spouse/partner, children, parents, etc. Four items were included that were originally part of the *Social Support at Work and Family Scale* [60] which originally has 36 particles. We selected only the items related to support from family members. Example of an item: "I can speak of my work to my family members without embarrassment." The respondent answers by indicating the level of agreement with each statement on a 7-point scale (from 1—I do not agree at all, to 7—I agree completely). The total score is obtained by adding the scores of the associated particles, with a higher score after recoding negatively worded particles reflecting a higher level of perceived social support. Cronbach $\alpha$ is 0.85.

*Family finances* were checked with item: "How satisfied are you with your family's financial situation?" Answers were given on a scale from 1 (completely dissatisfied) to 5 (completely satisfied).

The *job characteristics* were checked using the *Demand Support Control Questionnaire*, 17-item version [46], raised from the demand–control model of job stress proposed by Karasek [61]. Questionnaire includes the three scales: psycho-logical demands (five items), decision latitude (six items), and social support at work (six items). Participants were asked to report how often they experience described situations (for psychological demands and decision latitude) and to report their level of (dis)agreement (for social support at work) with statements. All estimates are given on a four-point Likert scale, with higher values indicating higher psychological demands (range 5–20), higher decision latitude (range 6–24), and higher social support at work (range 6–24). All scale scores were calculated by summing item scores after appropriate reverse scoring of two items (4—overtime work and 9—variety of work). Since there is currently no Croatian version of this questionnaire, the authors translated it and confirmed the expected three-factor structure with exploratory factor analysis. Cronbach $\alpha$ are 0.77 (demands), 0.74 (control), and 0.90 (support).

In order to assess *satisfaction with society*, more precisely satisfaction with its three systems (healthcare, social welfare, and educational), a questionnaire designed for study purposes was used. The content of the items is based on the statements of the parents of children with developmental disabilities who participated in in-depth interviews in the qualitative part of the research [13], which preceded this quantitative study. The questionnaire contains 15 questions, 5 for each system, related to the knowledge and experience of professionals working in the system, their friendliness and interaction, clear information, prescribed laws, and their implementation. Participants indicate their satisfaction on a 5-point scale (1—completely dissatisfied; 5—completely satisfied). They also had the option of selecting the response "I have no experience/don't know." An average score is calculated

for each system, with a higher score corresponding to greater satisfaction. Cronbach α are 0.87 (healthcare system), 0.90 (social welfare system), and 0.92 (educational system).

Some characteristics were checked by the *sociodemographic part of the questionnaire*, such as the age, level of education, number of children, and the level of child's disability.

### *2.3. Analytical Strategies*

Since the aim of this study is to examine the extent to which life satisfaction of our target group of participants can be explained by factors from different domains, hierarchical regression analysis was used to analyze the results. Data were analyzed using the IMB Statistical Package for Social Sciences (SPSS, version 22; IBM Corporation, Armonk, NY, USA). One of the basic prerequisites for the implementation of regression analysis is that the distribution of the criterion variables, in our case life satisfaction, does not deviate significantly from the normal distribution. This criterion is met considering the low values of the index of skewness (−0.49; S.E. = 0.11) and kurtosis (−0.45; S.E. = 0.22). The second condition is that the predictor variables are significantly correlated with the criterion (except for the variables introduced in the first step as a kind of control) and that there is no high collinearity (≥0.7) between the predictor variables. This is checked by calculating bivariate correlations for study variables. The third condition refers to a low level of multicollinearity so that the results of the regression analysis can be interpretable. The obtained VIFs range from 1.01 to 1.94. All of the stated conditions are met.

In hierarchical regression analysis, five-step process was followed. In the first step, we entered age and level of education. The level of education was recorded so that there were two levels (1—lower level of education (primary and secondary school); 2—higher level of education; Bachelor's, Master's, and Ph.D. degrees) as control variables. In the following steps, variables from different domains were introduced in this order: personal, family, work, and social. This order was chosen according to the general principle that the more stable characteristics are included first in the regression analysis and only then those that are more prone to change.

### 3. Results

Table 2 shows the basic descriptive data of the predictor variables (except for the control variables, which are listed in Table 1) and the criterion variable. Considering the low values of the skewness and kurtosis, the distribution of the results does not deviate significantly from the normal distribution. Most of the variables have slightly above-average values, as indicated by the positive skewness. Exceptions are the variables of parental stress and the number of children.

**Table 2.** Descriptive parameters of the variables included in regression analysis.

| Domain | Variable | N | *M* | *SD* | Obtained Range | Theoretical Range | Skewness | Kurtosis |
|---|---|---|---|---|---|---|---|---|
| Personal | Optimism | 508 | 21.77 | 4.16 | 6–30 | 6–30 | −0.76 (0.11) | 1.12 (0.22) |
| | Personal Strength | 508 | 13.32 | 4.34 | 0–20 | 0–30 | −0.96 (0.11) | 0.97 (0.22) |
| Family | Number of children | 508 | 2.19 | 0.98 | 1–6 | 1-N/A | 1.10 (0.11) | 1.85 (0.22) |
| | Parental stress | 508 | 42.76 | 10.53 | 19–79 | 18–90 | 0.49 (0.11) | 0.32 (0.22) |
| | Family finances—satisfaction | 508 | 3.27 | 0.92 | 1–5 | 1–5 | −0.36 (0.11) | 0.04 (0.22) |
| | Social support (family members) | 504 | 21.03 | 5.11 | 5–28 | 4–28 | −0.63 (0.11) | −0.22 (0.22) |
| Work | Demands | 506 | 13.23 | 2.80 | 5–20 | 5–20 | −0.08 (0.11) | −0.27 (0.22) |
| | Control | 505 | 17.84 | 2.89 | 8–24 | 6–24 | −0.49 (0.11) | 0.01 (0.22) |
| | Support | 506 | 18.59 | 3.98 | 6–24 | 6–24 | −0.75 (0.11) | 0.46 (0.22) |
| Society satisfac-tion | Healthcare sy. | 504 | 2.93 | 0.87 | 1–5 | 1–5 | −0.15 (0.11) | −0.44 (0.22) |
| | Social welfare sy. | 464 | 2.93 | 0.98 | 1–5 | 1–5 | −0.17 (0.11) | −0.67 (0.23) |
| | Educational sy. | 475 | 3.09 | 1.01 | 1–5 | 1–5 | −0.24 (0.11) | −0.76 (0.23) |
| Criterion variable—life satisfaction | | 508 | 20.92 | 6.72 | 5–34 | 5–35 | −0.49 (0.11) | −0.45 (0.22) |

For each variable, the number of participants for whom data are available is also listed. The N is not the same for all variables, i.e., most of the data are "missing" for the variables that relate to satisfaction with the educational and social welfare system. We cannot consider these data missing in a classical sense, since we have deliberately left open the possibility that parents do not have to evaluate these systems if they do not have experience with them and do not know how they function. Before conducting the correlational analyses, we did not choose the option of replacing missing data because this would reduce the variability of the data, which would be reflected in later analyses.

For that reason, correlation and hierarchical regression analysis included 429 participants and these results are shown in Tables 3 and 4. Prior to conducting this study, the minimum sample size was determined. Based on the parameters for regression analyses [62]: $p$ (alpha) = 0.05, power = 0.80, number of predictors = 14, the minimum sample size was set at N = 133.

Predictors from different domains are correlated with life satisfaction in the expected direction (Table 3). Values of the correlation coefficients indicate that there is a strong relationship between life satisfaction with optimism and parental stress, a moderate relationship with personal strength, family finances, social support from family members, support from work colleagues, satisfaction with the healthcare and welfare system, and weak relationship with the number of children, demand, control and satisfaction with the educational system.

The results of the regression analysis show that the control variables entered in the first step (age and educational level) predicted a significant proportion of the variation in life satisfaction, namely 3% ($R^2$ = 0.03, F (2/426) = 6.07, $p < 0.01$), whereas only age was a significant predictor. The negative direction of the beta coefficient means that younger mothers report higher life satisfaction. We did not include the severity of the child's disability as a control variable, although it has been recognized as a predictor of parents' well-being [26,27]. There are two reasons for that. Firstly, some of the parents (N = 91) do not know the degree of impairment because they did not go through the process of official classification of difficulties as defined by the Unique Body of Expertise Act [63] (2016), which would drastically decrease the number of participants in the correlational analyzes. On the other hand, in the subsample of parents who know the degree of impairment, the correlation between the degree of impairment and life satisfaction is not significant (r = −0.08, $p > 0.05$).

Table 3. Correlations for study variables (N = 429).

|  |  | 1 | 2 | 3 | 4 | 5 | 6 | 7 | 8 | 9 | 10 | 11 | 12 | 13 | 14 |
|---|---|---|---|---|---|---|---|---|---|---|---|---|---|---|---|
| 1. | LF |  |  |  |  |  |  |  |  |  |  |  |  |  |  |
| 2. | Age | −0.15 ** |  |  |  |  |  |  |  |  |  |  |  |  |  |
| 3. | ED | 0.05 | 0.10 * |  |  |  |  |  |  |  |  |  |  |  |  |
| 4. | OP | 0.54 ** | −0.09 | −0.01 |  |  |  |  |  |  |  |  |  |  |  |
| 5. | PSth | 0.40 ** | −0.04 | −0.07 | 0.50 ** |  |  |  |  |  |  |  |  |  |  |
| 6. | NC | 0.12 * | 0.13 ** | −0.21 ** | 0.15 ** | 0.08 |  |  |  |  |  |  |  |  |  |
| 7. | ParS | −0.61 ** | 0.18 ** | 0.13 ** | −0.47 ** | −0.46 ** | −0.14 ** |  |  |  |  |  |  |  |  |
| 8. | FF | 0.46 ** | −0.03 | 0.21 ** | 0.20 ** | 0.20 ** | −0.02 | −0.15 ** |  |  |  |  |  |  |  |
| 9. | SSF | 0.46 ** | −0.14 ** | 0.01 | 0.39 ** | 0.29 ** | 0.06 | −0.39 ** | 0.32 ** |  |  |  |  |  |  |
| 10. | D | −0.22 ** | 0.03 | 0.01 | −0.13 ** | −0.15 ** | −0.02 | 0.19 ** | −0.17 ** | −0.28 ** |  |  |  |  |  |
| 11. | C | 0.11 * | −0.06 | 0.35 ** | 0.20 ** | 0.07 | −0.07 | 0.02 | 0.18 ** | 0.17 ** | 0.03 |  |  |  |  |
| 12. | S | 0.30 ** | −0.11 * | −0.01 | 0.38 ** | 0.25 ** | 0.11 * | −0.26 ** | 0.22 ** | 0.44 ** | −0.36 ** | 0.33 ** |  |  |  |
| 13. | HCS | 0.36 ** | 0.06 | 0.04 | 0.18 ** | 0.13 ** | 0.06 | −0.23 ** | 0.26 ** | 0.26 ** | −0.14 ** | 0.06 | 0.22 ** |  |  |
| 14. | WFS | 0.33 ** | 0.01 | 0.08 | 0.14 ** | 0.12 * | 0.01 | −0.22 ** | 0.26 ** | 0.26 ** | −0.16 ** | 0.08 | 0.21 ** | 0.63 ** |  |
| 15. | EDS | 0.27 ** | −0.02 | 0.01 | 0.13 ** | 0.16 ** | 0.11 * | −0.27 ** | 0.16 ** | 0.23 ** | −0.10 * | 0.067 | 0.27 ** | 0.54 ** | 0.48 ** |

Note: * $p < 0.05$, ** $p < 0.01$; LF—life satisfaction; ED—level of education (1—lower level; 2—higher level); OP—optimism; PSth—personal strength; NC—number of children; ParS—parental stress; FF—family finances; SSF—social support from family members; D—demands; C—control; S—support (from work colleagues); HCS—healthcare system; WFS—welfare system; EDS—educational system.

**Table 4.** Model coefficients of hierarchical regression analysis of observed variables in predicting life satisfaction (N = 429).

| Predictors | 1st Step | 2nd Step | 3rd Step | 4th Step | 5th Step |
|---|---|---|---|---|---|
| | ß | ß | ß | ß | ß |
| Age | −0.161 ** | | | | |
| Education | 0.062 | | | | |
| Optimism | | 0.444 ** | | | |
| Personal strength | | 0.178 ** | | | |
| Number of children | | | 0.042 | | |
| Parental stress | | | −0.405 ** | | |
| Family finances | | | 0.304 ** | | |
| Social support from family members | | | 0.094 * | | |
| Demand | | | | −0.042 | |
| Control | | | | −0.005 | |
| Support | | | | −0.033 | |
| Healthcare system | | | | | 0.114 ** |
| Social welfare system | | | | | 0.058 |
| Educational system | | | | | −0.16 |
| $R^2$ | 0.03 | 0.33 | 0.58 | 0.58 | 0.60 |
| Adjusted $R^2$ | 0.02 | 0.33 | 0.57 | 0.57 | 0.58 |
| $\Delta R^2$ | 0.03 | 0.30 | 0.25 | 0.002 | 0.02 |

Note: * $p < 0.05$, ** $p < 0.01$.

Introducing individual characteristics significantly increased the percentage of explained variance in life satisfaction, reaching a level of 33% ($\Delta R^2$ = 0.30 F (2/424) = 97.63, $p < 0.01$). The beta coefficients indicate that both introduced variables (optimism and personal strength) contribute to such a significant increase. Higher levels of optimism and greater personal strength were associated with greater life satisfaction. In the third step, predictors from the family domain were introduced, including the number of children, parental stress, satisfaction with family finances, and social support from family members related to the work role. This significantly increased the percentage of explained variance in life satisfaction to 58% ($\Delta R^2$ = 0.25, F (4/420) = 61.042, $p < 0.01$). Examination of the t-scores of the beta coefficients showed that the number of children did not contribute to the increase in explained variance, while the other three variables did. Lower levels of parental stress, greater satisfaction with finances, and support from family members were associated with higher levels of life satisfaction. In the fourth step, factors from the work dimension were introduced: psychological demands, control, and social support at work. The entry of these three new variables did not significantly contribute to the percentage of variance explained, but it remained at the same level as after the third step ($\Delta R^2$ = 0.002, F (3/417) = 0.64, $p > 0.05$). In the last step, variables from the social domain were entered: satisfaction with the healthcare, welfare, and educational system. In this step, the percentage of explained variance in life satisfaction increased to 60%, i.e., this group of predictors brought an increase of 2% ($\Delta R^2$ = 0.02, F (3/414) = $p < 0.01$, where the beta coefficient was significant only for satisfaction with the healthcare system. Higher satisfaction with this system was predictive of higher life satisfaction. Other predictors entered in the earlier steps that remained significant in the final step of the regression analysis were optimism, parental stress, family finances, and social support from family members related to work.

## 4. Discussion

This study differs from most similar studies examining the well-being of parents of children with disabilities in that it comprehensively measures the number of predictors and the domains to which these predictors relate. We considered that this type of approach would allow us to obtain a clearer picture of the importance of each predictor as well as their

interrelationships. Therefore, the aim of this study was to determine the separate and joint contribution of individual, family, occupational, and societal characteristics in explaining the life satisfaction of working mothers whose children have developmental disabilities. Considering the raised hypotheses, we can say that most of them were confirmed, but some of the results are not consistent with them. By this, we primarily refer to the nonsignificant contribution of predictors from the work domain to the explanation of variance in life satisfaction, which will be discussed later.

Among control variables (age and education), age proved to be a significant predictor of life satisfaction, in the way that older participants reported lower life satisfaction, which is consistent with the results of other studies [64]. A longitudinal study [64] that collected data 10 years apart and included parents of children with and without disabilities, confirmed some previous findings but also led to some new ones. As in previous research [7–9], parents of children with disabilities were confirmed to have lower well-being compared to parents of children without disabilities. However, it was also found that the changes over the 10-year period were not the same for these two groups of parents. Parents of children with developmental disabilities had a greater decline in well-being, operationalized as mental health, over time than parents of typically developing children. Considering that our study included parents of children up to the age of 19 and that the average age of a child with disabilities is 9.2 years, we can say that our sample certainly includes parents who are dedicated to caring for their children over a longer period of time. Thus, the relationship between age and life satisfaction is probably due to the duration of exposure to the increased effort associated with caring for child/children with disabilities and the changes that result from this increased care.

Greater optimism and personal strength, as variables from the personal domain, were found to be predictors of greater life satisfaction, as expected. Their inclusion in the regression analysis increased the percentage of explained variance in life satisfaction by a substantial 30%, suggesting that these parental characteristics are very important to their well-being. Comparing optimism and personal strength, the results show that optimism is still a better predictor of life satisfaction. Such a result was also expected when considering what these two variables represent. Personal strength, as one of the dimensions of posttraumatic growth, is a subjective experience of coping with the consequences of a child's diagnosis. Optimism, as operationalized in this study, represents a general personality disposition [15]. Dispositional optimism is a personality trait that refers to generalized positive expectations about future outcomes in the broadest sense and is independent of the child's specific upbringing.

The results further showed that family characteristics were also significant in explaining the variance in life satisfaction, with the number of children not being a significant predictor, while the other three variables were. Parental stress proved to be the strongest predictor of life satisfaction. Since parental stress is perceived as a negative feeling toward oneself and the child/children that is directly attributed to the demands of parenthood [65], it is evident why there is a clear negative relationship between stress and life satisfaction. The importance of parental stress in predicting life satisfaction is related to the importance of parental role. In a study [13] in which employed parents of children with disabilities were interviewed, parents generally emphasized that their parental role came first and only then all other roles, including work role. The importance of financial resources was also confirmed in this study, which was expected given the findings of some previous research that caregivers of children with disabilities are more likely to experience most types of financial stress compared to caregivers of typically developing children [5] and that financial security is related to the well-being of children and their parents [6,29]. Given the increased demands associated with caring for a child with difficulties, parents welcome any form of support from their environment. By this, we mean informational, emotional, and instrumental support, which can come from a variety of sources. Since the participants in this study are working mothers, we were particularly interested in how much support they receive from family members to help them balance work and family roles. Balancing

these two roles is challenging for parents of children with normal development, and it is even more difficult for this group of parents [66]. As expected, this predictor also proved significant for their life satisfaction. After analyzing the importance of individual predictors from the family domain, it can be said that the dynamics of the family are characterized by financial resources, social support which family members provide each other, and the stress arising from the parental role, which plays a more important role in determining parental well-being from the fact for how many children they care.

Interestingly, the predictors from the work domain did not contribute to explaining the variance in life satisfaction, and such a result is not what was expected. Although significant correlations of job characteristics (job demands, control, support) with life satisfaction were found at the bivariate level in the expected directions (negative correlation with job demands and positive correlation with control and support), none of the predictors proved significant in the regression analysis. We assume that this result is due to the specificity of the sample. First of all, it is obvious that we included only a small percentage of mothers of children with developmental disabilities in the sample since the most recent data on the number of children with developmental disabilities in the Republic of Croatia indicate almost 70,000. Of course, not all mothers are employed, but our sample is certainly biased. Namely, participants in this study represent an extremely homogeneous and selected group of participants, which is even a rarity considering the data from the literature. In the situation of raising a child with difficulties, mothers usually withdraw from the labor market [6] (as fathers are typically perceived as breadwinners) and take over a large part of childcare. These mothers have not done so, although in most cases they are mothers whose children have severe and most severe impairments (Table 1). This suggests that these are mothers for whom it is extremely important to be fulfilled in their work role. Since the data presented in this study are only a part of the broader research, we had the opportunity to verify this assumption. Namely, when we compared the opinion of working and non-working mothers about the importance of realization through work and career, it was far more important for working mothers (F $(1/766)$ = 59.04, $p < 0.01$). Moreover, in explaining the results of work domain predictiveness, the fact that mothers, regardless of whether they work or not, usually take on more household and childcare responsibilities should be considered [67]. All these factors can explain why the results suggest that occupational characteristics are unimportant. They are only irrelevant in the context described, and the question arises as to what results would have been obtained if fathers had been included in the sample. Indeed, research suggests that mothers and fathers adapt differently to the role of parenting a child with disabilities [68].

In the last step of the regression analysis, variables from the social domain were introduced, representing access to necessary services and formal support that parents and their children can expect from society. Satisfaction with the functioning of the healthcare, educational, and welfare system significantly contributed to explaining life satisfaction (2%), although to a much lesser extent than the introduction of factors from the personal (30%) and family (25%) domains. The increase in the explained variance of life satisfaction is a consequence of satisfaction with the functioning of the healthcare system, considering that only this variable was a significant predictor. Parents' statements about the functioning of these three systems in the Republic of Croatia [13] indicate the importance of various programs and policies within all three systems. Parents highlighted as positive examples the stay of a child with disabilities in school or kindergarten, the right to an assistant, organized transportation to school, financial benefits, and the right to free healthcare. The results of this study suggest that the quality and quantity of care for children's physical and mental health are more important for parental well-being than some other rights. This finding is particularly valuable in the context of the results of some previous studies that highlighted the over-centralization of the healthcare system in the Republic of Croatia, which makes it difficult to access services in smaller communities, as well as the lack of a sufficient number of specialists in the public health sector, which parents compensate for by financing the necessary services in the private sector [69]. The insignificance of

satisfaction with the functioning of the educational and welfare system as predictors of life satisfaction should be taken with caution because some things can be compensated by parents themselves, e.g., they may compensate for increased financial needs with more working hours.

Some specific policy and research implications can be drawn from these results. How parents deal with the fact that their child's development does not follow the usual trajectory, and with everything that comes with caring for a child with developmental difficulties, depends largely on their individual characteristics. Therefore, the improvement of the well-being of parents could be made through different kinds of intervention programs. Since in this research, optimism and personal strength are recognized as predictors of well-being, such programs could be focused on parents' belief systems. This study is another in a series of studies that confirm the importance of social support as a protective factor for parental well-being. Therefore, different forms of education and workshops on this topic should be organized. In order to reduce parental stress, parents should be offered various forms of social support beyond what their own family members can provide them. Policy-makers in Croatia should also focus on better elaboration of the rules and conditions for financial relief to which families are entitled. The current reliefs available to parents are assessed as insufficient in most cases [13]. The quality of the healthcare system should definitely be considered, as satisfaction with the functioning of this system is important for parents of children with developmental disabilities. As for the factors related to the work domain, it would not be justified to draw any practical conclusions. Due to a number of shortcomings of this study (described in more detail later in the text), primarily related to sample bias and homogeneity, and the fact that fathers were not included in the sample, the results suggesting that predictors from the work domain do not contribute to explaining variance in life satisfaction have low ecological validity. To be able to draw some constructive conclusions, it would be necessary to conduct research on unbiased and heterogeneous samples of working parents of children with and without developmental disabilities. We believe that such research might succeed in identifying factors that are found to be protective of parents' well-being in some previous studies, such as work schedule flexibility, peer and supervisor support, workplace policies, and organizational culture [13,45].

Finally, we would like to point out some advantages and shortcomings of this study. The main advantage is comprehensiveness in terms of the number of predictors and the number of domains from which the predictors originate. Such research is truly rare, that is, most quantitative research on predictors of well-being for parents of children with disabilities is partial in nature, that is, it focuses on a smaller number of predictors. The study most similar to this study in terms of capturing a large number of predictors [66] used the bioecological model to examine work–family balance among working parents of children with disabilities, taking into account personal resources, personal demand, and microsystem (supervisor support), exosystem (organizational cultures), and macrosystem (national family support policies) variables. In addition, the advantage is that only parents whose children had an officially identified level of disability were included in the study.

A serious limitation of the study concerns the fact that it was conducted only on mothers and not on fathers. Thus, this study magnifies the difference between the number of studies conducted on mothers only and the number of studies that include both parents and fathers only. Although we originally intended to include both parents in the study, we faced the problem that not enough fathers completed the questionnaire, and we could not achieve that their share in the sample was at least 10%. Thus, we did not have the opportunity to compare mothers and fathers, and the previous literature shows that fathers' adjustment processes to parenting children with difficulties differ significantly from those of mothers [68]. This is particularly important in the context of studying working parents, as fathers are typically perceived as breadwinners [36] and therefore their work role is less threatened due to having a child with developmental disabilities. As mentioned earlier, the question arises whether the predictors from the work domain would be insignificant

if we had fathers in the sample for whom the work role is more important compared to mothers' perception of this role. One of the limitations is related to the fact that the participants filled in the questionnaire voluntarily, without any compensation, so it is possible that only those mothers who were intrinsically motivated to say something about the problems related to raising children with disabilities entered the survey, which also limits the possibility of generalizing the results obtained. Another factor that could affect the homogenization of the sample is the bias in data collection resulting from the fact that the data were collected online. This likely resulted in recruiting the portion of that part of respondents who are more skilled in the use of IT technology, and this may be correlated with some sociodemographic variables (e.g., education). Of course, all these factors seriously limit the possibility of generalizing the obtained results.

## 5. Conclusions

Hierarchical regression analysis indicated that factors from the personal domain (optimism and personal strength), family domain (satisfaction with family finances, parental stress, and support from family members related to work), and society domain (satisfaction with the healthcare system) predicted parents' life satisfaction. The total amount of explained variance in life satisfaction is a high 60%. The predictors from the work domain were not found to be significant. A suggestion for future research relates to the inclusion of fathers in the sample, which would improve the possibility of generalizing the data.

**Author Contributions:** Conceptualization, M.N.I. and A.S.; methodology, M.N.I., A.S., J.O., A.T. and T.B.; formal analysis, M.N.I.; writing—original draft preparation, M.N.I.; writing—review and editing, A.S., J.O., A.T. and T.B. All authors have read and agreed to the published version of the manuscript.

**Funding:** This research was funded by University of Zadar, Croatia. The results presented in this paper represent only a portion of the findings obtained as part of the University approved and funded project "Well-being of employed parents of children with developmental disabilities" (IP.01.2021.16).

**Institutional Review Board Statement:** The study was conducted in accordance with the Declaration of Helsinki, and the protocol was approved by Ethics Committee of the X (Klasa: 602-04/21-01/12; Urbroj: 2198-1-79-41/21-01; 12 July 2021).

**Informed Consent Statement:** Informed consent was obtained from all subjects involved in the study.

**Data Availability Statement:** The data are available on request from the authors of the study.

**Acknowledgments:** A special thanks to all the participants who took the time to complete our questionnaire. We also thank all those who helped with data collection.

**Conflicts of Interest:** The authors declare no conflict of interest.

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
