# Peer review of "Life Satisfaction in Employed Mothers of Children with Disabilities: The Importance of Personal, Family, Work, and Society Characteristics"

_societies, doi:10.3390/soc13080177_

Round 1

Reviewer 1 Report

I recommend explaining the Croatian context more in the introduction or in the discussion. For example, data on the number of employed mothers of children with disabilities (especially with severe difficulties) compared to the total number of mothers/children with disabilities. Challenges in accessing support, availability, differences, and limitations of support services in terms of regional differences or differences in terms of Zagreb and the rest of Croatia that contribute to mothers being employed or not and might affect  their feeling of life satisfaction. In addition, suggest how the results of this research could contribute to practice. For future research, it would be interesting to compare the life satisfaction of employed and unemployed parents of children with disabilities and, in this context, suggest some guidelines for ensuring and supporting parents' well-being.

Author Response

Dear Reviewer 1,

we thank you for the thorough reading of the article and giving the constructive suggestions to improve its quality. Below you will find your suggestions and answers in detail.

  1. I recommend explaining the Croatian context more in the introduction or in the discussion. For example, data on the number of employed mothers of children with disabilities (especially with severe difficulties) compared to the total number of mothers/children with disabilities.

In the introduction we added next text under the subtitle Family domain: „Data on the prevalence of children with developmental disabilities vary depending on the method of data collection, the definition used, and cultural differences in the understand-ing of developmental disabilities [21]. According to the International Classification of Functioning, Disability and Health for Children and Youth [22], difficulties in child de-velopment include four main domains: body structures (e.g., the structure of the nervous, sensory, cardiovascular, and other body systems), body functions (e.g., sensory, mental, psychomotor, etc.), limitations in activities (e.g., walking, reading, learning) and participa-tion (e.g., playing), as well as physical, social, and psychological environmental factors. The mentioned areas are also included in the Croatian national classification of children with developmental disabilities: "A child with developmental disabilities is a child who, due to physical, sensory, communication, language or intellectual disabilities, needs ad-ditional support for learning and development in order to achieve the best possible de-velopmental outcome and social inclusion" [23] (p. 10). Estimates at the global level [24] suggest that there are between 2.5 and 10% children with developmental disabilities in the population. According to the data from the Croatian National Register of Persons with Disabilities [25], the number of children whose developmental difficulties are classified as disabilities (age 0-19) is 69,953, of which 42,910 are boys and 27,043 are girls, and their share in the general population of children is 9.4%. “

Unfortunately, the national report on persons with disabilities (Benjak, 2022) does not include data on the percentage of children aged 0 to 19 years in categories of different severity of impairment. These data are only available at a general level that includes all people with disabilities, but not by different age categories. We also do not know what percentage of the mothers of these children are employed.

Furthermore, the Croatian context is additionally explained in the part related to the rights that parents and their children with disabilities can exercise.
One part was added to the introduction, under subtitle Society

„ The rights that parents and their children with developmental disabilities can exercise re-late to three broad groups: the family's right to financial support (the right to personal disability allowance, the right to assistance and care allowance, the right to compensation for education-related transportation costs, the right to children's allowance), the parents' rights under the parental work system (the right to part-time work, the right to caregiver status), and the children's rights under the educational and rehabilitation system (the right to psychosocial support, early intervention services, the right to assistance with inte-gration into educational programs) [50].”.

The other part (related to croatian context) is added in disussion in section where results of regression analysis related to predictorst from  social domain are explained: „ This finding is particularly valuable in the context of the results of some previous studies that highlighted the over-centralization of the health care system in the Republic of Croa-tia, which makes it difficult to access services in smaller communities, as well as the lack of a sufficient number of specialists in the public health sector, which parents compensate for by financing the necessary services in the private sector [69]. “

Since this text refers to the regional differences in the exercise of rights, it is also a response to your second comment.

  1. Challenges in accessing support, availability, differences, and limitations of support services in terms of regional differences or differences in terms of Zagreb and the rest of Croatia that contribute to mothers being employed or not and might affect their feeling of life satisfaction.

Your suggestion about regional differences in service availability was accepted and explained in the previous comment.

  1. In addition, suggest how the results of this research could contribute to practice. For future research, it would be interesting to compare the life satisfaction of employed and unemployed parents of children with disabilities and, in this context, suggest some guidelines for ensuring and supporting parents' well-being.

In the discussion, we added a text that refers to practical implications (related to the individual, family and social domain ), but also to implications for future research (related to the work domain). This part of the text is positioned after the interpretation of the results and before the description of the advantages and shortcomings of this study.

“Some specific policy and research implications can be drawn from these results. How parents deal with the fact that their child's development does not follow the usual trajec-tory, and with everything that comes with caring for a child with developmental difficul-ties, depends largely on their individual characteristics. Therefore, the improvement of well-being of parents, could be made through different kind of intervention programs. Since in this research, optimism and personal strength are recognized as predictors of well-being, such programs could be focused on parents’ belief systems. This study is an-other in a series of studies that confirm the importance of social support as a protective factor for parental well-being. Therefore, different forms of education and workshops on this topic should be organized. In order to reduce parental stress, parents should be of-fered various forms of social support beyond what their own family members can provide them. Policy makers in Croatia should also focus on better elaboration of the rules and conditions for financial relief to which families are entitled. The current reliefs available to parents are assessed as insufficient in most cases [13]. The quality of the health care sys-tem should definitely be considered, as satisfaction with the functioning of this system is important for parents of children with developmental disabilities. As for the factors relat-ed to the work domain, it would not be justified to draw any practical conclusions. Due to a number of shortcomings of this study (described in more detail later in the text), primar-ily related to sample bias and homogeneity, and the fact that fathers were not included in the sample, the results suggesting that predictors from the work domain do not contribute to explaining variance in life satisfaction have low ecological validity. To be able to draw some constructive conclusions, it would be necessary to conduct research on unbiased and heterogeneous samples of working parents of children with and without develop-mental disabilities. We believe that such research might succeed in identifying factors that are found to be protective of parents' well-being in some previous studies, such as work schedule flexibility, peer and supervisor support, workplace policies, and organizational culture [13,45].”

Reviewer 2 Report

Thanks for inviting me to review this interesting and innovative manuscript. My  comments are included below, with detailed suggestions on how to improve this  manuscript, if the author(s) wish to revise it.

Title

The title clearly indicates that the study is about life satisfaction in employed mothers of children with disabilities, providing a specific and targeted subject. However, the following disadvantages are obvious.

(1) While it mentions personal, family, work and society characteristics broadly as important factors for life satisfaction prediction, it does not provide specific details or examples within each domain.

(2) The reader might need more information about what kind disability their child has, as different disabilities can have different impacts on parents' well-being.

Abstract

It highlights the importance of considering multiple domains when studying the well-being of parents with children with disabilities.Besides,the inclusion criteria for participants are clearly stated.

(1) However, the sample size is not mentioned in relation to statistical analysis or generalizability. There is no mention of limitations or potential biases in data collection or analysis methods used.

(2) It does not provide specific details about how personal, family, and societal factors were measured.

Overall, while the abstract effectively summarizes key aspects of the study's purpose and findings, it could be improved by providing more information on sample size and limitations. Additionally, including more details about measurement methods would enhance clarity.

Introduction

The introduction section of the text discusses the challenges faced by parents of children with developmental disabilities and the impact on their well-being. It highlights the similarities among parents in terms of long-term care, financial costs, and adjustments required for the child's development.

The following areas should be improved:

(1) The introduction could benefit from providing more specific information about the impact on parental well-being.

(2) While it mentions variability in parents' adjustment, it does not elaborate further on this point. Providing examples or discussing factors that contribute to this variability would enhance understanding.

Materials and Methods

There is no mention of any measures taken to ensure data quality or reliability. Including information on data validation procedures would strengthen this section.

Results

(1) This passage does not provide specific numerical values or statistical tests used to analyze data.

(2) Besides, It does not discuss any significant findings or relationships between variables.

Discussion

The following ideas should be considered:

(1) Lack of explanation for unexpected results: While the discussion mentions unexpected results such as nonsignificant contribution from predictors in the work domain, it does not provide an explanation or explore potential reasons behind these findings.

(2)The paper focuses specifically on working mothers with children with developmental disabilities and does not discuss how these findings may apply to other populations or contexts.

(3)The author should consider discussing the practical implications of their findings. How can this knowledge be used to develop interventions or provide support for mothers of children with disabilities? This would enhance the relevance and applicability of the study's results.

no

Author Response

Dear Reviewer 2,

we thank you for the critical evaluation of our article and the suggestions, which we have taken into account to a greater extent, thus achieving greater clarity and quality of the work. In the following, we address your comments point by point.

  1. Title

The title clearly indicates that the study is about life satisfaction in employed mothers of children with disabilities, providing a specific and targeted subject. However, the following disadvantages are obvious.

(1) While it mentions personal, family, work and society characteristics broadly as important factors for life satisfaction prediction, it does not provide specific details or examples within each domain.

(2) The reader might need more information about what kind disability their child has, as different disabilities can have different impacts on parents' well-being.

Regarding the comment related to providing specific details or examples of predictors within each domain in the title of the paper, we propose not to make these changes, i.e., we thin  it is better that the title in the revised version of the article includes only four domains of predictors. We justify this by the fact that in this artile we used tvelve predictors from four different domains. To list them all would result in a too long (and unattractive) title. If we were to pick out some of predictors according to a particular principle, we do not know what that principle should be. Certainly not the significance of the predictors, because we consider the results showing insignifiant beta coefficients for predictors that we expect to be significant to be very valuable.

From the accompanying list of references, it is evident that we used a relatively large number of sources in the conceptualization and writing of this article. A general rule we came across in reviewing the literature is that the title indicates what type of disability child has, if it is a homogeneous group of parents and/or children in terms of this criterion. For example, Al-Kuwari M. Psychological health of mothers caring for mentally disabled children in Qatar. Neurosciences. 2007;12(4):312-7. On the other hand, papers that use a heterogeneous sample of parents and/or children in relation to diagnosis usually do not mention any of them in the title itself. For example, Byra S, Żyta A, Ćwirynkało K. Posttraumatic growth in mothers of children with disabilities. Croatian Rev For Rehabil Research. 2017;53,Supplement:15-27.
We also believe that emphasizing the heterogeneity of the sample in terms of the types of disabilities (e.g., by adding the term "parents of children with different types of disabilities") would lead readers to the wrong conclusion that the paper presents results related to different categories / types of disabilities, which is not the case.

  1. Abstract

It highlights the importance of considering multiple domains when studying the well-being of parents with children with disabilities.Besides,the inclusion criteria for participants are clearly stated.

(1) However, the sample size is not mentioned in relation to statistical analysis or generalizability. There is no mention of limitations or potential biases in data collection or analysis methods used.

(2) It does not provide specific details about how personal, family, and societal factors were measured.

Overall, while the abstract effectively summarizes key aspects of the study's purpose and findings, it could be improved by providing more information on sample size and limitations. Additionally, including more details about measurement methods would enhance clarity.

According to the suggestion to mention sample size, the following sentence was added to the Results section: „ Prior to conducting this study, the minimum sample size was determined. Based on the parameters for regression analyses [62]: p (alpha) = 0.05, power = 0.80, number of predic-tors = 14, the minimum sample size was set at N=133.“

In the discussion (part that refers to interpreting the results related to predictors from the work domain), a sentence was added referring to the possibility of generalizing the results.„First of all, it is obvious that we included only a small percentage of mothers of children with developmental disabilities in the sample, since the most recent data on the number of children with developmental disabilities in the Republic of Croatia indicate almost 70,000. Of course, not all mothers are employed, but our sample is certainly biased.“ Data releted to number of children with disabilities are also aded in the introduction, in accordance with the comments of the first reviewer.

In the discussion, in the section describing the shortcomings of the investigation, the following text was added, referring to biases in data collection: „Another factor that could affect the homogenization of the sample is the bias in data col-lection resulting from the fact that the data were collected online. This likely resulted in recruiting the portion of that part of respondents who are more skilled in the use of IT technology, and this may be correlated with some sociodemographic variables (e.g., education).“

Considering the limitation regarding the length of the abstract (the abstract should be a total of about 200 words maximum ), the corrections in accordance with the suggestions were incorporated into the abstract as much as was allowed due to the propositions. We added a key sentence in the abstract related to generalizability: "Considering the sample size and bias, these results have significant limitations in terms of generalizability." It is also emphasized what factors for each domain were measured by online questionnaire (“They completed an online questionnaire to measure factors from personal (optimism and personal strength), family (satisfaction with family finances, parental stress, number of children and support from family members related to work), work (job demands, control, and support) and society domain (satisfaction with the health care, educational and welfare system). To comply with instructions to authors regarding the length of the abstract, we removed some parts: first sentence and " .. distributed primarily through institutions that serve children with disabilities and their parents."

  1. Introduction

The introduction section of the text discusses the challenges faced by parents of children with developmental disabilities and the impact on their well-being. It highlights the similarities among parents in terms of long-term care, financial costs, and adjustments required for the child's development.

The following areas should be improved:

(1) The introduction could benefit from providing more specific information about the impact on parental well-being.

(2) While it mentions variability in parents' adjustment, it does not elaborate further on this point. Providing examples or discussing factors that contribute to this variability would enhance understanding.

In the part of the introduction that refers to the work domain, the following part about well-being was added: „ The relationship between the working status of parents and the well-being of parents of children with disabilities is not unambiguously determined. For some parents of children with disabilities, employment may play a protective role in well-being because of the many functions that work has, but it may also be an additional source of stress that can lead to lower well-being [7].“

Each of the factors described under the Individual domain, family domain, work domain and soiety contributes to variability of adaptation and well-being. This part was obviously not clear enough, so the sentence from the previous version of the artile is extended; „Obviously, in this "journey" from finding out the child's diagnosis to well-being, there are many factors that can have a protective or aggravating effect, and thus contribute to the variability of adjustment and well-being“.

  1. Materials and Methods

There is no mention of any measures taken to ensure data quality or reliability. Including information on data validation procedures would strengthen this section.

We note that for each measurement instrument there is information on its internal type reliability, expressed by the Cronbach alpha coefficient.

Moreover, in the description of each instrument there is a reference showing that almost all the questionnaires used (those related to life satisfation, optimism, personal strength, parental stress, social support for family members) imply the use of adapted and validated versions in Croatian.

The only exception is our self-translated Demand Suport Control Questionnaire, which belongs to the category of very commonly used questionnaires and has been validated in several countries. We confirmed its three-factor structure with an exploratory factor analysis. This section has now been added to the text of the description of this measurement instrument: “Since there is currently no Croatian version of this questionnaire, the authors translated it and confirmed the expected three-factor structure with an exploratory factor analysis.”

  1. Results

(1) This passage does not provide specific numerical values or statistical tests used to analyze data.

(2) Besides, It does not discuss any significant findings or relationships between variables.

The descriptive data are presented in Table 1, the bivariate correlations in Table 3, and the results of the hierarchical regression analysis in Table 4. Table 3 provides information on the significance of each correlation (p< 0.05 and p<0,01). In Table 4, the beta coefficients (with the corresponding significance levels) are presented, as well as R2, Adjusted R2 and ΔR2 for eah step. since this is the usual way of presenting these analyses, we do not know exactly which numerical values you mean? In accordance with the examples of some papers previously published in the same journal, we decided to present the data analysis in two sections: Analytial Strategies and Results. This way seemed appropriate because in the first part we described general things related to the analysis (preconditions for the analysis, normality of the distribution), and in the second part we dealt with the results of the analysis performed.

After Table 3 we have added a sentence referring to the values of the correlation coefficients between the predictor variables and the criterion variable: „Values of the correlation coefficients indicate that there is a strong relationship of life sat-isfaction with optimism and parental stress, moderate relationship with personal strength, family finances, social support from family members, support from work col-leagues, satisfaction with healthcare and welfare system, and weak relationship with number of children, demand, control and satisfaction with educational system.“

  1. Discussion

The following ideas should be considered:

(1) Lack of explanation for unexpected results: While the discussion mentions unexpected results such as nonsignificant contribution from predictors in the work domain, it does not provide an explanation or explore potential reasons behind these findings.

(2)The paper focuses specifically on working mothers with children with developmental disabilities and does not discuss how these findings may apply to other populations or contexts.

(3)The author should consider discussing the practical implications of their findings. How can this knowledge be used to develop interventions or provide support for mothers of children with disabilities? This would enhance the relevance and applicability of the study's results.

In the part referring to unexpected results, we have added text: „ First of all, it is obvious that we included only a small percentage of mothers of children with developmental disabilities in the sample, since the most recent data on the number of children with developmental disabilities in the Republic of Croatia indicate almost 70,000. Of course, not all mothers are employed, but our sample is certainly biased.“

This text actually fits with what was mentioned earlier as a possible reason that the predictors from work domain did not contribute to the percentage of explanation of variance in life satisfaction.

These results are explained in context:
1. the homogeneity of the sample, which is related to findings from the existing literature (mothers tend to withdraw from the labor market as fathers are typically perceived as breadwinners). Apart from having in common that they are employed, most of them are mothers caring for children with severe and most severe impairments, which further contributes to increasing homogenization.
2. the importance of realization through work

when we compared the opinion of working and non-working mothers on this topic, we confirmed this assumption. Results of ANOVA are presented in discussion.

  1. that mothers tend to take on more responsibilities in the household and in childcare
    For this reason, it can be expected that their life satisfaction is determined to a greater extent by other factors, which do not include the work domain.

We hope that this part of the results, together with the changes already mentioned, is now well explained.

Related to your second and third comment we added a text that refers to practical implications (related to the individual, family and social domain), but also to implications for future research (related to the work domain). This part of the text is positioned after the interpretation of the results and before the description of the advantages and shortcomings of this study.

“Some specific policy and research implications can be drawn from these results. How parents deal with the fact that their child's development does not follow the usual trajectory, and with everything that comes with caring for a child with developmental difficulties, depends largely on their individual characteristics. Therefore, the improvement of well-being of parents, could be made through different kind of intervention programs. Since in this research, optimism and personal strength are recognized as predictors of well-being, such programs could be focused on parents’ belief systems. This study is another in a series of studies that confirm the importance of social support as a protective factor for parental well-being. Therefore, different forms of education and workshops on this topic should be organized. In order to reduce parental stress, parents should be offered various forms of social support beyond what their own family members can provide them. Policy makers in Croatia should also focus on better elaboration of the rules and conditions for financial relief to which families are entitled. The current reliefs available to parents are assessed as insufficient in most cases [13]. The quality of the health care system should definitely be considered, as satisfaction with the functioning of this system is important for parents of children with developmental disabilities. As for the factors related to the work domain, it would not be justified to draw any practical conclusions. Due to a number of shortcomings of this study (described in more detail later in the text), primarily related to sample bias and homogeneity, and the fact that fathers were not included in the sample, the results suggesting that predictors from the work domain do not contribute to explaining variance in life satisfaction have low ecological validity. To be able to draw some constructive conclusions, it would be necessary to conduct research on unbiased and heterogeneous samples of working parents of children with and without develop-mental disabilities. We believe that such research might succeed in identifying factors that are found to be protective of parents' well-being in some previous studies, such as work schedule flexibility, peer and supervisor support, workplace policies, and organizational culture [13,45].”

Round 2

Reviewer 2 Report

  • Thanks to the author for the revision. I think this manuscript is accepted

  • Thanks to the author for the revision. I think this manuscript is accepted